# Genes Involved in Immune Reinduction May Constitute Biomarkers of Response for Metastatic Melanoma Patients Treated with Targeted Therapy

**DOI:** 10.3390/biomedicines10020284

**Published:** 2022-01-26

**Authors:** Miguel-Angel Berciano-Guerrero, Rocío Lavado-Valenzuela, Aurelio Moya, Luis delaCruz-Merino, Fátima Toscano, Javier Valdivia, Victoria Castellón, Fernando Henao-Carrasco, Pilar Sancho, Juan-Luis Onieva-Zafra, Ismael Navas-Delgado, Antonio Rueda-Dominguez, Elisabeth Perez-Ruiz, Emilio Alba

**Affiliations:** 1Medical Oncology, IBIMA, Medical Oncology Intercenter Unit, Hospital Regional Universitario de Malaga, Regional and Virgen de la Victoria University Hospitals, 29010 Málaga, Spain; juanluonieva@gmail.com (J.-L.O.-Z.); rueda.dominguez@gmail.com (A.R.-D.); ELIPERU@GMAIL.COM (E.P.-R.); ealbac@uma.es (E.A.); 2Clinical and Translational Cancer Research Group, IBIMA, 29010 Málaga, Spain; rocio.lavado@ibima.eu; 3Cancer Molecular Biology Lab, 29010 Málaga, Spain; amoyag@uma.es; 4Medicine School, University of Málaga, 29010 Málaga, Spain; 5Medical Oncology, Virgen Macarena University Hospital, 41009 Seville, Spain; luis.cruz.sspa@juntadeandalucia.es (L.d.-M.); ferheca@gmail.com (F.H.-C.); 6Medical Oncology, Hospital Juan Ramon Jimenez, 21005 Huelva, Spain; fatimeska@hotmail.com; 7Service of Medical Oncology, Hospital Universitario Virgen de las Nieves, 18014 Granada, Spain; jvaldib@gmail.com; 8Medical Oncology, Complejo Hospitalario Torrecardenas, 04009 Almería, Spain; victo_eu@yahoo.es; 9Medical Oncology, Hospital Universitario Virgen del Rocio, 41013 Seville, Spain; sanchomarquez@gmail.com; 10Khaos Research, University of Malaga, 29010 Málaga, Spain; ismael@uma.es

**Keywords:** melanoma, targeted therapy, biomarker, immunology

## Abstract

Targeted therapy in metastatic melanoma often achieves a major tumour regression response and significant long-term survival via the release of antigens that reinduce immunocompetence. The biomarkers thus activated may guide the prediction of response, but this association and its mechanism have yet to be established. Blood samples were collected from nineteen consecutive patients with metastatic melanoma before, during, and after treatment with targeted therapy. Differential gene expression analysis was performed, which identified the genes involved in the treatment, both in the first evaluation of response and during progression. Although clinical characteristics of the patients were poorer than those obtained in pivotal studies, radiological responses were similar to those reported previously (objective response rate: 73.7%). In the first tumour assessment, the expression of some genes increased (CXCL-10, SERPING1, PDL1, and PDL2), while that of others decreased (ARG1, IL18R1, IL18RAP, IL1R1, ILR2, FLT3, SLC11A1, CD163, and S100A12). The analysis of gene expression in blood shows that some are activated and others inhibited by targeted therapy. This response pattern may provide biomarkers of the immune reinduction response, which could be used to study potential combination treatments. Nevertheless, further studies are needed to validate these results.

## 1. Introduction

One of the functions of the immune system is to detect and eliminate cells presenting uncontrolled proliferation (i.e., tumour cells). In a healthy individual, the immune system must frequently control tumour cells that have formed spontaneously to prevent their settlement, progression and metastasis [1]. All of these processes are part of tumour immunology. The immune system plays a crucial role in combating melanoma, for example, and immunotherapy has been implemented in this understanding via immune checkpoint inhibitors. In this respect, the programmed death-1 (PD1) and cytotoxic T-lymphocyte associated antigen-4 (CTLA4) immune checkpoints are major resources against metastatic melanoma [2,3,4], as a standard treatment and in the adjuvant setting [5,6,7]. The publication of the Cancer Genome Atlas revealed many fruitful areas for treatment development and revolutionised the molecular biology of cancer. A recent advance in this respect is the introduction of targeted therapy to address driver mutations in various tumours [8], such as the BRAFV600 mutation, which occurs in 40–50% of patients with melanoma [9]. The development of BRAF inhibitors, together with the incorporation of MEK inhibitors (the main resistance mechanism), has made targeted therapy the standard treatment for patients with BRAF-mutated melanoma both in metastatic disease [10,11,12] and in the adjuvant scenario [13]. The choice between immunotherapy and targeted therapy is not always clear, and patient-selection data are often lacking. Although the involvement of BRAF in the immune system is well known, further study of this question is needed [14]. In recent years, studies have reported evidence, albeit patchy, on the relations between the two approaches: targeted therapy and tumour immunology [15]. Targeted therapy increases the expression of melanoma differentiation antigens, reduces levels of immunosuppressive cytokines in the microenvironment and enhances the CD8 T cell response and T-cell-mediated cytotoxicity. PDL-1 overexpression by melanoma cells has been identified, together with increased markers of immune depletion, including PD-1 and TIM-1, suggesting that the immune response is inhibited before resistance occurs. However, there has been less progress towards understanding how targeted therapy directly aimed at the immune environment of melanoma works. The development of appropriate biomarkers would improve our understanding of this question. The working hypothesis of this study, therefore, is that the use of BRAF inhibitors in patients with metastatic melanoma presenting a BRAF mutation provokes the reinduction of acquired immunity against the tumour and that changes in immunological blood biomarkers indicate this re-establishment.

## 2. Materials and Methods

### 2.1. Objectives and Endpoints

PRIMARY ENDPOINTS:-Identify changes in immunological markers after administering targeted therapy (BRAF+/−MEK inhibitors in patients with a BRAF mutation) whose mechanisms of action can be correlated with good immune function. In this sense, we will consider immune reinduction.-Identify early response prediction markers to identify long-term survivors after treatment with targeted therapy.-Identify prognostic immunological markers of metastatic melanoma.

SECONDARY ENDPOINTS:-Correlate the blood gene expression of immunological markers and their monitoring after the treatments with clinical variables.-Generate hypotheses to optimise the sequence of treatments (targeted therapy vs. immunotherapy).

### 2.2. Patients and Study Design

This prospective study was conducted using blood samples collected from a sample of patients with metastatic melanoma treated with BRAF inhibitors with or without MEK inhibitors (REINMEME study). The inclusion criteria were that the participants should have BRAF-mutated metastatic melanoma, be at least 18 years old and have no previous diagnosis of cancer. They could have received adjuvant treatment for early melanoma but never with BRAF-MEK inhibitor or anti-PD1 therapy. The patients were seen in the hospital once monthly, and the first tumour assessment was conducted at 10–14 weeks. The RECIST v.1.1 criteria were applied for response evaluation [16], and the patients were grouped as follows: very poor responder if progression-free survival (PFS) was less than 4 months and overall survival (OS) less than 9 months, poor responder if PFS was 4–6 months or OS was 9–12 months, good responder if PFS was 6–10 months or OS was 12–24 months and very good responder if PFS was >10 months or OS was >24 months.

### 2.3. Ethical Aspects

All patients provided written informed consent to take part in this study, which was conducted in accordance with the Declaration of the Helsinki World Medical Association and was approved by the local Ethics Committee in July 2015.

### 2.4. Sample Collection

Blood samples were collected before starting treatment (T0), at the first radiological evaluation (T1) and during progression (T2). In patients who failed to progress after 18 months of targeted therapy, a third sample was also extracted (T2). Patients who, for medical or other reasons, did not provide the three samples or who presented other circumstances that might alter the results were excluded.

### 2.5. Gene Expression Analysis

Gene expression analysis was performed using peripheral blood samples anti-clotted with EDTA. The RNA was extracted and purified using the Quiagen RNA Amp Minikit in accordance with the manufacturer’s protocol. The concentration and purity of the genetic material were quantified in Nanodrop One. Samples that did not meet the required quality standards were discarded. Gene expression analysis was performed on the Nanostring nCounter system, using the nCounter PanCancer Immune Profiling PanelTM. This panel measures the expression levels of 770 genes that include markers of 24 cell types and immune populations, 30 common antigens against cancer and genes that represent all categories of the immune response, including key blocking genes. The quality of the data obtained in nCounter was checked using nSolver Analysis Software 3.0 (Nanostring). This software was also used to normalise the data and to perform the necessary quality controls to ensure there had been no problem during the analysis and that the results obtained for the tumour were consistent with the reality. Only those samples that met the quality standard were included in the differential expression and co-expression analyses, which were performed with the DESeq2 library in R v.4.1 [17].

### 2.6. Statistical and Bioinformatic Analysis

A descriptive analysis of the study variables was carried out to determine the mean values, standard deviation or medians according to the distribution (symmetric or otherwise) and range (maximum, minimum). The categorical variables are presented as absolute and relative frequencies. Continuous quantitative variables, such as the number of copies of mRNA (in the gene expression measurement) in each sample, were compared using the Pearson linear correlation coefficient when the variables presented a normal distribution and the Spearman coefficient otherwise. Values above 0.6 were considered to represent a good positive correlation, and (arbitrarily, but in line with the literature) values below −0.6 were taken to indicate a negative correlation. We also calculated the corresponding significance of this coefficient to determine whether the value obtained showed that the variables were actually related or only presented this relationship by chance. Gene expression changes were calculated on a logarithmic scale (in base 2), and changes were considered significant for results <−1 or >1. Finally, statistical significance was assumed at *p* < 0.05. Dichotomous categorical independent variables (such as certain clinical variables) were compared with polychotomous categorical dependent variables (such as certain clinical variables or the measurement of protein expression by immunohistochemistry in paraffin tumour tissue) by the chi-square test or by Fisher’s exact test if the expected proportion of values <5 was greater than 20%. The survival analysis was carried out using the Kaplan–Meier method, applying the log rank test to the factors expected to present differences in the univariate statistical analysis and that from the clinical standpoint were both plausible and meaningful. Computer analysis of gene expression data was performed using semi-automatic model reconstruction methods based on multi-objective optimisation heuristic techniques. Additionally, to facilitate data interpretation, we designed VIGLA-M, a web service that allows clinicians to explore gene-expression data [18]. To carry out the experiments at the necessary scale, Big Data Analysis techniques were applied, using the computational resources of the Ada Byron Research Building at the University of Malaga. The co-expression of two genes is the correlation of their expression profiles across the set of samples studied. The expression profile of a set of genes can also be correlated with a clinical variable that describes a phenotypical feature of the samples. In this case, the genes are said to be involved in producing that particular feature. In the present study, we performed a co-expression analysis with WGNA [19] after the treatment (T1) to find genes associated with the clinical variables Responder, OS1, OS and PFS. Further functional analysis of these genes was performed with the R package ClusterProfiler [20] on Gene Ontology [21] and KEGG [22].

## 3. Results

### 3.1. Study Population

Between October 2015 and September 2019, twenty-six patients gave signed consent to participate in the study, but only nineteen were fully studied (Appendix A). Some of the characteristics of these patients, including tumours, blood tests and the treatments received, are shown in Table 1. The mean age at diagnosis of metastatic disease was 49.9 years (range 23.9 to 85.4). The mean time from the diagnosis of melanoma to the diagnosis of metastatic disease was 26.2 months (range 0–102), although 38.9% of patients debuted as stage IV. A total of 63.2% were female. Most presented an adequate performance status (ECOG 0–1 in 78.39%). At the start of the targeted therapy, 73.7% had M1c-d according to the TNM Classification of Malignant Tumours, 8th edition (26.3% metastases in the central nervous system), and 47.1% presented high levels of lactate dehydrogenase (LDH). All patients except one received combined therapy with BRAF and MEK inhibitors. The remaining patient had a contraindication and started with BRAF inhibitor monotherapy. Tolerance to the medication was acceptable, with grade 3–4 toxicity of 36.8%, which required a 52.6% dose adjustment.

### 3.2. Treatment Response and Survival

The mean objective response rate (ORR) was 73.7%. A total of 21% of the patients were classified as very poor responders, 16% were poor responders, 26% were good responders and 37% were very good responders. The median overall survival (OS) from the start of targeted therapy was 10.7 months (95% CI 8.8–12.6) and the median progression-free survival (PFS) was 9.3 months (95% CI 4.9–13.9) (Figure 1a,b). The univariate analysis of overall survival by subgroups is summarised in Appendix A. Good performance status was associated with a good prognosis. Statistically significant differences were found for adjuvant treatment but were contrary to expectations. Moreover, the patients with high LDH or leukocytosis had a worse prognosis and might constitute a subgroup to which a non-standard strategy should be applied (see Appendix A). The multivariate analysis of survival rates, using Cox regression, did not reveal a significant association between response to treatment and the other clinical variables analysed (see Appendix A).

### 3.3. Gene Expression

From the 770 genes in the panel, we selected for analysis those with a statistically significant differential expression between T0 (before starting treatment) and T1 (at the first radiological assessment). This analysis showed that the expression of some genes increased (CXCL10, SERPING1, PD-L1 or PD-L2) and others decreased (ARG1, IL18R1, IL18RAP, IL1R1, ILR2, FLT3, SLC11A1, CD163 or S100A12) (see Table 2). These genes participated in most of the immunological functions studied, both directly and indirectly. We then considered how they evolved to progression, that is, between T1 and T2. Analysis showed that in the progression, the genes that had been overexpressed presented a down-regulation, while those that had been under-expressed had an up-regulation (see Figure 2). We also examined which genes were modified between T0 and T2, finding statistically significant changes only in NFKBIA. However, the evolution from the first time point was downward, so this change cannot be considered a prognostic biomarker. In our analysis of the co-expression of genes, we looked for genes that had a similar expression pattern and relationship with different clinical variables (ECOG, metastasectomy, adjuvant treatment or LDH stratification) or with the response to treatment (OS, PFS or responder/non-responder) at T0 and T1. Those meeting these parameters, together with the module genes, are shown in the Appendix A. It is worth highlighting the genes of the green module in T1 (see Appendix A) since they are significantly related to response and survival (overall survival and progression-free survival). In addition, a survival analysis was performed based on the baseline expression of these genes (see Figure 3).

## 4. Discussion

To our knowledge, this paper is the first to show that the study of immune reinduction evidenced in peripheral blood could help to predict the response to targeted therapy treatments. To date, no routine biomarkers have been approved to determine early response in patients with metastatic melanoma treated with BRAF-MEK inhibitors. Some studies have shown that the serial determination of the BRAF mutation in circulating tumour DNA (ctDNA) can help predict the response to treatments for patients with melanoma [23,24,25], but this technique has very limited implementation. In addition, studies have been undertaken to carry out genomic analyses to predict survival or treatment response in mutated BRAF melanoma, but these studies all used tumour tissue and, in most cases, few samples were taken, due to the risks involved in rebiopsy, among other considerations [26]. The co-expression analysis carried out in our work shows genes that are mainly involved in the activation of the innate immune response (GO: 0045089 and GO: 0002218), in the NF-kappa beta signalling pathway (KEGG hsa04064) and in the production of cytokines (GO: 0001819). Several recent studies have enhanced our understanding of the relationship between BRAF and MEK inhibition and the response of the immune system, both acquired [14] and innate [27,28]. These studies have shown that the combination of BRAF and MEK inhibitors has immunological effects, reducing cell populations and immunosuppressive cytokines and improving the expression and expansion of immune-system-stimulating molecules and cytokines. Most of these studies have been performed in vitro or in vivo [29]. However, various clinical trials have also been undertaken to evaluate the combination of BRAF-MEK inhibitors with immunotherapy based on immune checkpoints, and so we expect to receive more data shortly [30]. In this paper, we analysed the state of the immune system, in real time, via a minimally invasive approach aimed at determining the expression of immune genes involved in the response to targeted therapy. Among other findings, we report an increase in the expression of PD-L1, PD-L2, CXCL10 and SERPING1 related to the initial radiological response. The expression of PD-L1 in tissue does not seem to be a good biomarker in metastatic melanoma treated with targeted therapy [31], although its expression does increase in the microenvironment after the inhibition of BRAF, both in preclinical models and in tumour biopsies [32,33]. To our knowledge, no prior studies have been undertaken to determine PD-L1/PD-L2 in blood and to correlate their presence with the response to treatment for metastatic melanoma, although comparable studies have been conducted regarding other tumours [34,35,36]. Some papers have described the relation between PD-L2 expression and the presence of lymphatic metastases [37]. In addition, Kakavand et al. [38] described a series of cases where immune activity (based on RNA expression matrices) was studied in the biopsies of patients with BRAF-mutated melanomas and PD-L1 expression. Although transcriptomic signatures indicative of immune cell activation were found to be correlated with PD-L1 staining, this association persisted in biopsies performed following tumour progression. Unfortunately, this means that PD-L1 expression cannot be used as a marker of response or resistance. By contrast, our study observed a reduction in PD-L1/PD-L2 expression with tumour progression, which is indicative of a possible mechanism of acquired resistance to Immune Checkpoint Inhibitors (ICI). As with PD-L1/PD-L2, we detected an increased expression of CXCL-10, a chemokine related to the trafficking of immune cells and which presents reduced expression with progression. In this respect, Dufour et al. studied animal models deficient in CXCL-10 and observed trafficking alterations, mainly of Th1 cells, and a lower generation of antigen-specific effector T cells [39]. This anti-tumour property has attracted considerable research attention, especially in the field of melanoma [40]. For example, Barreira da Silva et al. conducted an in vitro study using dipetidylpeptidase 4 inhibitors and observed an increase in CXCL-10 levels and a suppression of tumour growth in melanoma models [41]. Our study also detected an increase in SERPING-1, which encodes C1INH, an inhibitor of the classical complement activation pathway. Although its correlation with drug response has not been fully established, studies have reported complement-mediated cytotoxicity in metastatic melanoma [42]. Our work also shows that the response to combined BRAF-MEK therapy is associated with a decreased expression of CD163, ARG1, IL1R or IL18R, among others. Regarding CD163, a recent study reported that the activation of effector T cells and the depletion of macrophages 163+ in the tumour are associated with a good response to BRAF inhibitor treatment in patients with metastatic melanoma [43]. Arginase is an enzyme that is mainly present in neutrophils, although it can also be seen in regulatory T lymphocytes and granulocytic-myeloid-derived suppressor cells (G-MDSC) [44]. ARG1+ macrophage depletion has been related to the response to anti-PD-1 therapy in murine models [45], although to our knowledge, it has not been associated with the response to BRAF-MEK inhibitors. Furthermore, differential expression analyses have measured a significant reduction in the expression of inflammatory interleukin receptors. BRAF inhibition is known to activate the inflammasome and increase IL-1 production in dendritic cells [46,47], improving the activity of caspase-1 and increasing the cellular response of T lymphocytes [48]. IL1R1, the IL-1 beta receptor, reduces the recognition of melanocytic cells by the immune system. According to in vitro studies by Zubrilov et al. [49], vemurafenib-resistant cell lines exhibit an increased expression of IL1R1 mRNA. The same response was observed in the present study with tumour progression. Our study has both strengths and weaknesses. A limitation is the small sample size considered, which means the conclusions drawn are not especially robust. However, our findings are in line with the literature in this respect, which leads us to believe that a study of biomarkers in peripheral blood through NGS platforms is a viable approach. Furthermore, the nCounter platform is not accessible to the entire scientific community, which may limit the development of a possible biomarker. However, many centres have developed this technique, and the constitution of a network would facilitate a larger sample size and possible validation. In addition, our sample is characterised by a profile of especially young patients with a disease onset that offers a poor prognosis (due to the high percentage of elevated LDH and brain metastases observed). This group of patients responds poorly both to targeted therapies and to immunotherapy [50,51]. However, this circumstance also lends our study greater clinical interest since there is currently a dearth of treatments for this subgroup of patients. In summary, our findings reflect the changes associated with immune reinduction caused by targeted therapy in patients with disseminated melanoma, which had been previously described for in vitro models and in paraffin tissues. Furthermore, our work describes resistance mechanisms such as the increase in IL1R1 to progression, detected in a dynamic and minimally invasive way. The dynamic behaviour of the immune system means that further study, of a prospective and serial nature, is needed to confirm our findings.

## 5. Conclusions

In patients with metastatic melanoma with a BRAF mutation, BRAF/MEK inhibitors can modify the immune profile. The analysis of blood samples by next-generation sequencing, focusing on genes related to the immune system, would enhance the prediction of response to treatment.

## 6. Patents

The results of this study have been communicated, and a patent has been applied for, registered in May 2020 at the Junta de Andalucía (Spain) as number 2020/45. This process caused an unavoidable delay in the publication of the present article.

## Figures and Tables

**Figure 1 biomedicines-10-00284-f001:**
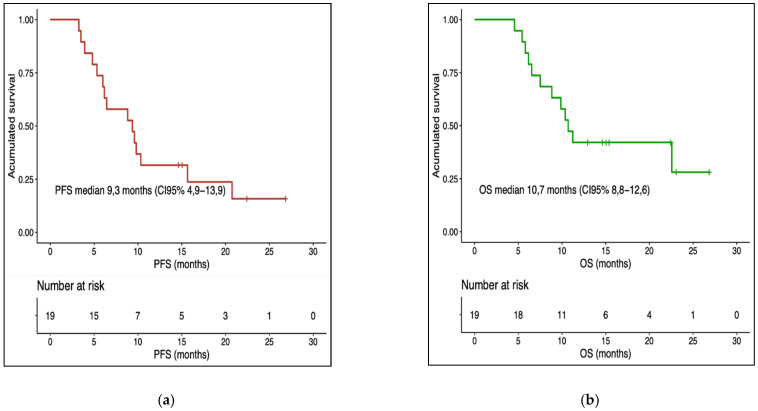
Kaplan–Meier. (**a**) Progression-free survival analysis. (**b**) Overall survival analysis.

**Figure 2 biomedicines-10-00284-f002:**
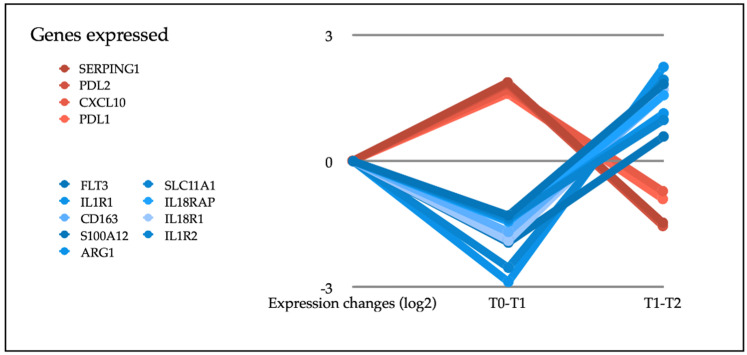
Expression changes produced by targeted therapy.

**Figure 3 biomedicines-10-00284-f003:**
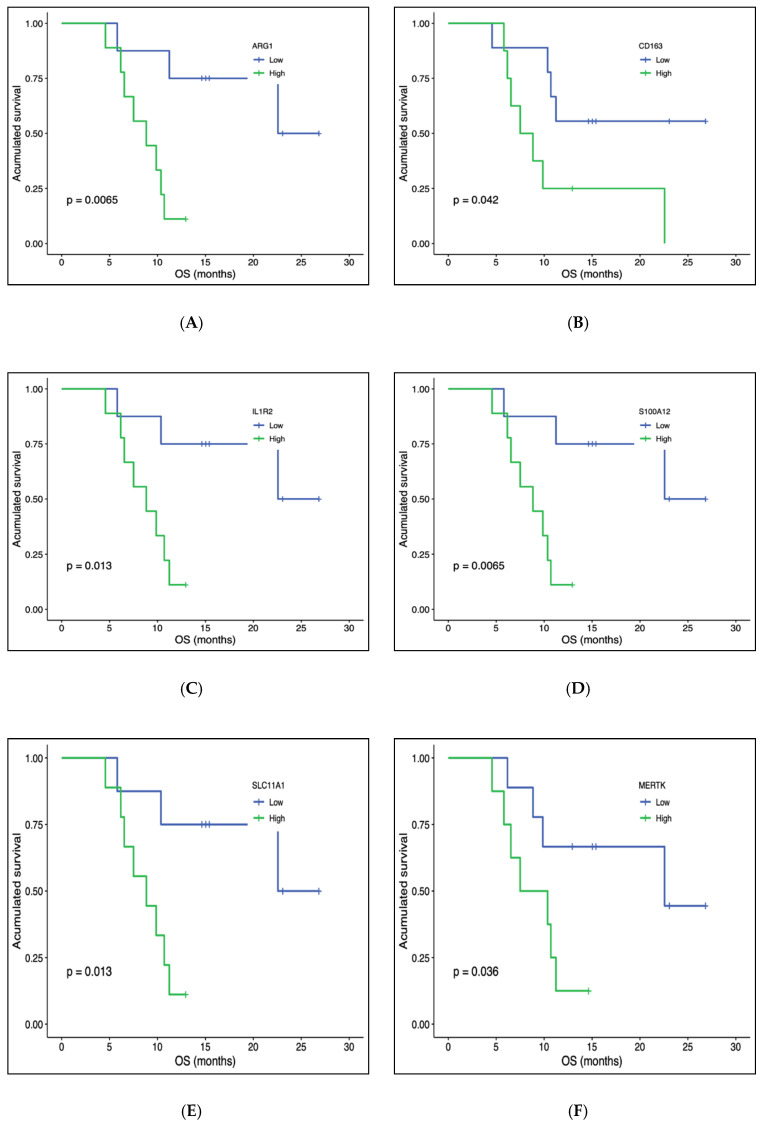
Kaplan–Meier overall survival analysis by baseline gene expression (only statistically significant genes are represented). (**A**) ARG1. (**B**) CD163. (**C**) IL1R2. (**D**) S100A12. (**E**) SLC11A1. (**F**) MERTK.

**Table 1 biomedicines-10-00284-t001:** Baseline characteristics of patients.

Patient Characteristics	N = 19
Age at initial diagnosis (mean, range)	45.7 (20.7–61.9)
Age at stage IV diagnosis	49.9 (23.9–85.4)
Male	36.8%
Female	63.2%
PS 0–1	78.9%
PS 2	21.1%
Comorbidities	
Allergies	15.8%
Other medical conditions	5.3%
**Baseline Conditions**	**N = 19**
Previous treatments	
Primary tumour resection	77.8%
Metastasectomy	33.3%
Adjuvant	27.8%
Analytics	
LDH (mean, range)	367UI (144–1350)
Elevated LDH	47.1%
Lymphocytes (mean, range)	1485.3 (500–2600)
High dNLR	68.4%
**Tumour features**	**N = 19**
Initial stage	
Stage I-II	44.4%
Stage III	16.7%
Stage IV	38.9%
Ulceration	38.5 %
Primary tumour location	
Limbs	27.8%
Trunk	33.3%
Head and neck	22.2%
Special locations *	16.7%
Months Dx primary-M1 (mean, range)	26.2 (0–106)
Number of metastasis	
One	21.1%
Two	26.3%
Three or more	52.6%
CNS metastasis	26.3%
M1a-b	26.3%
M1c-d	73.7%
BRAF Mutation	
V600E	94.7%
V600K	5.3%
**Treatment and response**	**N = 19**
Type of treatment	
Vemurafenib-Cobimetinib	57.9%
Dabrafenib-Trametinib	36.8%
iBRAF monotherapy	5.3%
Dose reduction	52.6 %
Toxicity	
No toxicity	5.3%
Mild–moderate toxicity (G1–2)	57.9%
Significant toxicity (G3–4)	36.8%
Response to treatments	
Disease stabilisation	21.1%
Partial response	57.9%
Complete response	15.8%
Progression	5.3%
Events	
Progression	78.9%
Death	63.2%
Survival (median in months, 95% CI)	
PFS	9.3 (4.8–13.9)
OS	10.7 (8.7–12.6)
OS (global)	12.2 (4.9–19.5)

* 3 patients unknown primary, 1 patient mucosal melanoma.

**Table 2 biomedicines-10-00284-t002:** Differential gene expression analysis (T0–T1).

Gene.Name	log2Fold Change	Wald Test	*p* Value	*p* Adjusted
SERPING1	1.8682	4.3611	0.0000	0.0020
PDCD1LG2 (PD−L2)	1.8222	5.0262	0.0000	0.0002
CXCL10	1.6894	3.8960	0.0001	0.0084
CD274 (PD−L1)	1.5948	4.5412	0.0000	0.0011
FLT3	−1.2846	−3.4812	0.0005	0.0320
SLC11A1	−1.3549	−3.3898	0.0007	0.0414
IL1R1	−1.4037	−3.7701	0.0002	0.0114
IL18RAP	−1.4360	−3.8318	0.0001	0.0098
CD163	−1.6765	−4.2324	0.0000	0.0030
IL18R1	−1.8855	−4.5545	0.0000	0.0011
S100A12	−1.9305	−4.1486	0.0000	0.0035
IL1R2	−2.5257	−4.1305	0.0000	0.0035
ARG1	−2.8609	−5.0351	0.0000	0.0002

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
