# Peer review of "Genes Involved in Immune Reinduction May Constitute Biomarkers of Response for Metastatic Melanoma Patients Treated with Targeted Therapy"

_biomedicines, 2022, doi:10.3390/biomedicines10020284_

Round 1

Reviewer 1 Report

This manuscript performed gene expression analysis using Nanostring to discover novel prognostic (OS and PFS) or predictive (response to targeted treatment with kinase inhibitors) genomic biomarkers in 19 melanoma patient blood samples.

There are several major and minor concerns that should be addressed, which are detailed below.

Major concerns:

1) The manuscript figure (2 figures) and table (1 table) number are below average for a research article, and typical of a brief report instead.

2) There is missing (title and abstract) or discrepant (materials and methods, results, discussion, and conclusions) information regarding which targeted treatment was administered to patients throughout the manuscript. The materials and methods indicate BRAF inhibition alone as a primary endpoint, and BRAF+MEK inhibition in the patient and study design (although 1 patient received BRAF inhibitor alone). The discussion and conclusions only refer to BRAF+ MEK inhibition, does this mean that the 1 patient who received BRAF inhibitor alone was excluded from the subsequent analysis?

3) Figures 1 (page 5) and 2 (page 7) are displaced in the manuscript and separated from their legends (page 7).

4) Description of the main findings resulting from the differential gene co-expression analysis (Tables S4 and S5) is missing in the results and discussion.

Minor concerns:

ABSTRACT

5) Replace the word “massive” with a more suitable scientific adjective.

6) Detail which targeted therapy was used.

INTRODUCTION

7) Page 1, 1st paragraph: Cancer immunology is not a term that describes how the immune system controls cancer progression, it is simply the study of the immune system in cancer. Revise this sentence.

8) Page 1, 1st paragraph: Remove the word “treatment” after “immunotherapy”.

9) Page 1-2: Kinase inhibitors should be described before immunotherapies, as this article is focused on the former.

10) Page 2, 3rd paragraph: Edit “targeted therapy and cancer immunology” to “targeted therapy and immunotherapy”.

11) Page 2, 3rd paragraph, “suggesting that the immune response is inhibited before resistance occurs.”: Isn’t immune regulation a form of resistance?

12) Page 2, 3rd paragraph, “However, there has been less progress towards understanding how therapy directly aimed at the immune environment of melanoma works”: this is not entirely correct, as there have been extensive studies on this topic in melanoma.

MATERIALS AND METHODS

13) Page 2, section 2.1: BRAF inhibitors or BRAF and MEK inhibitors?

14)  Page 2, section 2.1: Edit “long survivors” to long-term survivors”.

15) Page 2, section 2.1: Isn’t the sequence of treatment (targeted or immunotherapy) dependent on the current standard of care? In some countries, if patients are BRAF mutant, it is compulsory to receive kinase inhibitors upfront.

16) Page 2, section 2.2: Include which prior adjuvant treatments were permitted.

17) Page 3, section 2.3: Include the ethics number/identifier.

18) Page 3, section 2.4: How were T2 samples distinguished, as these include patients that progressed and those that didn’t? Were these samples separated in subsequent analyses?

RESULTS

19) All main and supplementary Tables and Figures: Abbreviations are not explained. Also, legends only include titles and lack descriptive information.

20) Page 4, section 3.2: Include how ORR was defined (includes partial and complete responses only?).

21) Page 5, section 3.2: It is rather strange that patients who received prior adjuvant treatment had lower OS when compared to those who didn’t. Did this include several treatments, and if so which ones? This should be mentioned in the discussion as well.

22) Figure 1: Figure and legend are incorrectly placed in the manuscript and should be on page 5 immediately after mention in the text. Graphs are not labeled as 1a or 1b. CI intervals of both graphs (4.9-13.9 and 8.8-12.6) do not match the text.

23) Table 1: Does T1 (1st radiological assessment) include responders and non-responders? If so, do these genes behave differently or the same for each response category? This will have implications on the meaning of these findings. Also, include the alias gene names for PD-L1 and PD-L2.

24) Figure 2: Figure and legend are incorrectly placed in the manuscript and should be on page 6 immediately after mention in the text. Does T2 include progressors and non-progressors? If so, do these genes behave differently or the same for each progression category? This will also have implications on the meaning of these findings. Different tones of red and blue are hard to distinguish. What about ARG1, IL18R1, ILR2, and S100A12?

25) Page 6, last paragraph: Include the main findings resulting from the differential gene co-expression analysis.

DISCUSSION

26) Page 7, 1st and 2nd paragraph: there are several other studies that have investigated on-treatment genomic changes in melanoma, and several biomarkers have been proposed, albeit not used in standard clinical practice. Revise these statements accordingly.

27) Page 8, 4th paragraph: ICI abbreviation is not explained. Why is resistance to ICI mentioned here, if these patients were treated with kinase inhibitors? Do authors mean that low PD-L1/2 expression would affect subsequent response to ICI? Needs clarification.

28) Page 8, last paragraph: MDSC-G abbreviation is not explained.

CONCLUSIONS

29) The manuscript does not clearly indicate how these data can be used to predict response to kinase inhibitors. Predictive biomarkers are present at baseline before treatment, while these data make use of three time points, and “predictive” genes are determined based on changes between T0 and T1 (10 to 14 weeks after treatment start).

SUPPLEMENTARY MATERIALS

30) Page 9: Table S5c is incorrectly mentioned as S6c.

31) Figure and table titles should be removed, and formatting should follow what was used in the main manuscript.

32) Figure S1: “non-consent” should be revised to “withdrawal of consent”. Description of censored patients between T0 and T1 does not add up to 7, but rather 5.

33) Figure S7: p-value in the legend (0.037) does not match Table S2 (0.051).

34) Table S1: The term “basal” should be replaced with “baseline”. “Special locations” should be described in the legend. “Metastasis location” should be replaced with “Number of metastasis”, and CNS metastasis should be separate. Type of treatment should not include “/” as this usually indicates one or the other, rather than a combination of both; and %s do not add up to 100% or seem correct (e.g. 1/19 = 5.26%, not 5.6%).

35) Table S2: Indicate significance in bold. Univariate analysis of OS and treatment response is not significant here (p-value=0.198), but Fig S3 comparing OS and response is significant (<0.001). Explain this discrepancy. What does “*” indicate in TNM stage?

36) Tables S4 and S5: It is unclear how module colors are defined and how to interpret these data tables. The meaning of these findings is unclear.

Author Response

Thanks to the reviewer for many of the considerations, which have improved the article, being more suitable for publication. If any correction or explanation is considered, do not hesitate to communicate it to us.

Reviewer 2 Report

Authors described changes associated with immune re-induction
caused by targeted therapy in melanoma patients. The article should be implemented, before to be accepted for the pubblication.

  1. Please add in the introduction consideration about the targeting of the ICOS/ICOSL system for melanoma therapy.
  2. Please provide the RIN values for the the RNA analyzed samples.
  3. Please show all the 770 analyzed genes by a Volcano plot to compare them before and after therapy.
  4. Please used Venn diagramm to shown genes shared by before and after therapies.
  5. Please, shows the possible biological networks to summurize the 770 genes (ie. by using Ingenuity pathway analysis).
  6. Did the authors confirmed the modulation of the selected genes by real time PCR? Please, provide data.
  7. Figure 2 is not visible.

MINOR

Replace cancer immunology with tumor immunology alonf the text

Insert features'patients in the main text.

Author Response

Thanks to the reviewer for their comments. Much of the requested content will be in the unpublished material, although we have no objection to including it in the manuscript or supplemental material if the reviewer finds it interesting to do so.

Round 2

Reviewer 1 Report

I thank the authors for addressing my concerns and editing the manuscript accordingly. 

I have no further comments.

Author Response

Different previously non-published graphs have been relocated in the latest version and added to the main manuscript in order to improve the presentation of results.

The manuscript has undergone English language editing by MDPI for this new version. 

Thank you for your comments and we look forward to the acceptance of the paper.

Reviewer 2 Report

The authors improved the manuscript by providing an answer for each comment, using the tools and data available to them. It may be of interest to include part of the non-published material in the main text, expecially for data with significant differences. The introduction should be implemented since inhibitors of PD1, PDL1 and CTLA4 are not the only ICPs.

Author Response

Thank the reviewer for his advice on data location. Different previously non-published graphs have been relocated in the latest version and added to the main manuscript.

Moreover, the manuscript has undergone English language editing by MDPI for this new version. 

Regarding the inclusion in the introduction of aspects related to ICP, our work is not focused on ICP, but on targeted therapy. The mention made of ICP ("In this respect, the programmed death-1 (PD1) and cytotoxic T-lymphocyte associated antigen-4 (CTLA4) immune checkpoints are major resources against metastatic melanoma") is to contextualize the approved treatments with immunotherapy in melanoma, which are exclusively anti-PD1 therapy and anti-CTLA-4 therapy. Indeed, there are many other ICPs with long development (anti-LAG-3 therapy, i.e.) and importance, but we consider that it would move away from the contextualization of our work. 

Thank you for your comments and we look forward to the acceptance of the paper.

This manuscript is a resubmission of an earlier submission. The following is a list of the peer review reports and author responses from that submission.

Round 1

Reviewer 1 Report

This is a biostatistical review.

  1. Please provide a summary of objectives and endpoints as a separate subsection in Section 2.
  2. Please provide a sample size and power justification.
  3. Please provide the corresponding protocol.
  4. Please provide the patient baseline characteristics as a table.
  5. In Table 1, what does ‘stat’?
  6. All Kaplan-Meier survival curves (including Figs. 1 and 2 and supplementary figures) should have the ‘number at risk’ with a proper label for the x-axis that indicates the start date of the duration.
  7. Please provide HR and 95% confidence intervals for Supplement Figs S2 to S7.
  8. In Tables S1-S3, please report HR and 95% confidence intervals.
  9. Did you check the proportional hazard assumption for log-rank tests and Cox regression? If so, did you see any violation? If so, how did you resolve it?

Author Response

Q1. Please provide a summary of objectives and endpoints as a separate subsection in Section 2.
This text was added to the main manuscript:
2.1. Objectives and endpoints
PRIMARY ENDPOINTS:
-Identify changes in immunological markers after the administration of targeted therapy (BRAF inhibitors in patients with a BRAF mutation) whose mechanisms of action can be correlated with good immune function. In this sense, we will consider immune reinduction.
- Identify early response prediction markers to identify long survivors after treatment with targeted therapy.
- Identify prognostic immunological markers of metastatic melanoma.
SECONDARY ENDPOINTS:
- Correlate the blood gene expression of immunological markers and their monitoring after the treatments, with clinical variables.
- Generate hypotheses to optimize the sequence of treatments (targeted therapy vs immunotherapy).

Q2. Please provide a sample size and power justification.
Initially, the sample size calculations were made taking as the reference population the Spanish population diagnosed with metastatic melanoma with BRAF V600E mutation, as well as assuming a confidence level of 95%, with a precision of 7% and a proportion of 5%. They were calculated with an expected percentage of losses of 15%. With these parametric values, a total loss-adjusted sample of 43 patients was calculated, for which a 12-month recruitment was estimated.
However, after a slow recruitment, mainly motivated by the geographic dispersion and the clinical aggressiveness of the disease, an amendment to the protocol was made and the T2 sample was introduced and the statistical power was recalculated, considering 19 patients sufficient to appreciate statistically significant variations in gene expression.

Q3. Please provide the corresponding protocol.
We can send the document, but in Spanish version.

Q4. Please provide the patient baseline characteristics as a table.
The table can be found in the supplementary material (S1). We had considered that it is too long a table for the main manuscript, but if the reviewer considers its inclusion in it, we would have no objection.

Q5. In Table 1, what does ‘stat’?
Wald statistic test. Table was modified.

Q6. All Kaplan-Meier survival curves (including Figs. 1 and 2 and supplementary figures) should have the ‘number at risk’ with a proper label for the x-axis that indicates the start date of the duration.
Figures were modified.

Q7. Please provide HR and 95% confidence intervals for Supplement Figs S2 to S7.
Figures were modified, but HR and confidence intervals are reflected in Table S2.

Q8. In Tables S1-S3, please report HR and 95% confidence intervals.
S1 is a descriptive table, so we consider that is not necessary, but tables S2-S3 were modified with these parameters.

Did you check the proportional hazard assumption for log-rank tests and Cox regression? If so, did you see any violation? If so, how did you resolve it?
After verifying that the model presented did not meet the Test the Proportional Hazards Assumption of a Cox Regression, it was modified so that said assumption was verified and we could give the model as valid.

English of the main manuscript has been reviewed by a native English professional. Any clarification do not hesitate to comment.

Reviewer 2 Report

I have some concerns about the manuscript that must be improved and answered 

  1. What do you mean with genes involved in immune reinduction? What immune reinduction means? Are the main functions of all the genes with differential expression described in the manuscript IMMUNE REINDUCTION? All of them could have immune associated functions, but I am not sure that reinduction is correct. Please check.
  2. In the title the authors conclude that the genes with differential  expression along target therapy could be biomarkers of response to the therapy, but no analysis in expression level versus response to therapy (very poor, poor, good responders) was done.
  3. Tables and Figures legends do not contain description at all. Figure 1 and 2 must be merged in just one figure. Supplementary figures and tables could be incorporated in the main manuscript. Figure 3 esthetic must be improved.

Author Response

Q1:

When we refer to the term immune reinduction, we are referring to a complex process, whose definition is not well established, and which defines the situation that would occur in cancer patients (whose immune editing mechanism is more defined and it is altered) than with treatment (in this case, targeted therapy) can reestablish proper immune function. Obviously, with this premise, immune reinduction is not a specific term that can be measured (there are many mechanisms involved), although it can be intuited through the different expression of genes involved in it. The greatness of this article is that it gives light to what happens in the expression of genes related to immunology, after an apparently non-immunological treatment. 
The inclusion of a section with objectives and endpoints, motivated by another reviewer, could clarify this aspect, but if the reviewer considers that we should make this definition more explicit, which in our opinion seems obvious, we would not have inconveniences.

Q2:

In the co-expression analysis we did include the type of responder, as can be seen in Figures S4a and S4b of the supplementary material. 

Q3:

Figures were modified. If the reviewer considers that a figure or table has to go in the main manuscript, we have no problem including it.

Round 2

Reviewer 1 Report

Overall, I am very concerned with the statistical power and analysis used in this manuscript, mainly because of the small sample size with unclear power/sample size justification. The provided sample size justification is not reproducible because several key components are missing. And the outcomes from Cox analyses do not make sense. For example, a categorical variable with three levels (e.g., ECOG) has only one HR and a categorical variable with three levels (e.g., previous treatment) has three HRs, which should have two HRs for both cases. The current multivariable outcomes are seriously underpowered because of too many covariates included with the small sample size of 19.

  1. What are the endpoints? Currently, only objectives are stated. Endpoints also should be stated clearly in the manuscript.
  2. The provided sample size and power statement are not clear, and several key components seem not included, such as endpoint, effect size, and targeted power. Because of these, it is almost impossible to reproduce the sample size and power justification. In addition, I cannot confirm whether 19 is enough due to a lack of information. I recommend providing the original sample size and power justification with the originally planned sample size (with clear endpoint, effect size, and targeted power/type I error rate). Then provide the revised sample size and power justification. Of course, these should be added to the manuscript.
  3. Provide the study protocol with the summary in English
  4. Add ‘N’ along with ‘%’ for categorical variables in Table S1
  5. Table S2 doesn’t make sense to me. Categorical variables must have a reference, but none has a reference. For example, for ‘Previous treatment’, all levels have HR without a reference. In addition, for the categorical variable ‘ECOG’, there are three levels, but only one HR is provided. What does ‘asterisk *’ mean? What does ‘not applicable’ mean?
  6. In Table S3, how did you choose these variables? What are the references? Most importantly, the current multivariable analysis seems seriously underpowered because of the small sample size of 19 with 7 covariates.
  7. It is not clear to me how the proportional hazard assumption was handled. Which variables were violated? What do you mean by ‘it was modified …’? There should be a statement for the remedy that was used.
  8. Replace ‘univariate’ and ‘multivariate’ with ‘univariable’ and ‘multivariable’, respectively.
  9. Due to the small sample size without formal sample size justification, this study should be considered an exploratory/proof-of-concept study. In this regard, the title must include the term ‘proof-of-concept’ or ‘exploratory study’.